# Novel Advances in Treatment of Meningiomas: Prognostic and Therapeutic Implications

**DOI:** 10.3390/cancers15184521

**Published:** 2023-09-12

**Authors:** Gerardo Caruso, Rosamaria Ferrarotto, Antonello Curcio, Luisa Metro, Francesco Pasqualetti, Paola Gaviani, Valeria Barresi, Filippo Flavio Angileri, Maria Caffo

**Affiliations:** 1Biomedical and Dental Sciences and Morphofunctional Imaging, Unit of Neurosurgery, University of Messina, 98122 Messina, Italy; rosamaria.ferrarotto@studenti.unime.it (R.F.); antonello.curcio@studenti.unime.it (A.C.); luisa.metro@studenti.unime.it (L.M.); fangileri@unime.it (F.F.A.); mcaffo@unime.it (M.C.); 2Radiotherapy Unit, Pisan University Hospital, 56100 Pisa, Italy; francep24@hotmail.com; 3Neuro Oncology Unit, IRCCS Foundation Carlo Besta Neurological Institute, 20133 Milan, Italy; paola.gaviani@istituto-besta.it; 4Department of Diagnostics and Public Health, Section of Pathology, University of Verona, 37134 Verona, Italy; valeriabarresi@univr.it

**Keywords:** biomarkers, genetics, meningioma, molecular therapy, surgery

## Abstract

**Simple Summary:**

Meningiomas are considered benign lesions and are frequently linked to a lower quality of life. Total surgical resection is the gold standard treatment. Combination treatments that target several molecular targets are becoming available and show great potential as adjuvant treatment alternatives. New classifications of these malignancies and novel therapeutic strategies are now possible thanks to recent developments in genetics, epigenetics, and, specifically, in the identification of specific genetic alterations. Although the outcomes to this point have not been very encouraging, different molecular-focused therapies have undoubtedly attracted a great deal of attention. Recent research has shown that microRNAs may have a role in the biology of meningiomas, allowing them to be used in meningioma treatment plans in the future.

**Abstract:**

Meningiomas are the most frequent histotypes of tumors of the central nervous system. Their incidence is approximately 35% of all primary brain tumors. Although they have the status of benign lesions, meningiomas are often associated with a decreased quality of life due to focal neurological deficits that may be related. The optimal treatment is total resection. Histological grading is the most important prognostic factor. Recently, molecular alterations have been identified that are specifically related to particular phenotypes and, probably, are also responsible for grading, site, and prognostic trend. Meningiomas recur in 10–25% of cases. In these cases, and in patients with atypical or anaplastic meningiomas, the methods of approach are relatively insufficient. To date, data on the molecular biology, genetics, and epigenetics of meningiomas are insufficient. To achieve an optimal treatment strategy, it is necessary to identify the mechanisms that regulate tumor formation and progression. Combination therapies affecting multiple molecular targets are currently opening up and have significant promise as adjuvant therapeutic options. We review the most recent literature to identify studies investigating recent therapeutic treatments recently used for meningiomas.

## 1. Introduction

Meningiomas are the most frequent histotype of tumors of the central nervous system (CNS). Their incidence is approximately 35% of all primary brain tumors and approximately 50% of all benign cerebral tumors [1]. These neoplasms are classified into three grades according to the World Health Organization (WHO) [2]. Commonly, patients with meningioma may present with headache, onset of critical episodes, altered mental status, speech impairment, strength impairment, and cranial nerve deficits. The neurological signs are clearly related to the site of the lesion. Initial treatment involves observation and surgical treatment, while radiotherapy and radiosurgery should be considered in cases of atypical and anaplastic meningioma. The site of the lesion as well as close proximity to vital structures can cause complications during surgical procedures. The treatment of recurrent tumors includes radiotherapy and repeated surgery. Some histotypes, despite surgical removal, show an aggressive trend with an early tendency toward recurrence. In these cases, radiation therapy can be used as adjuvant or complementary therapy. To date, the efficacy of chemotherapy and new molecular therapies is not yet clear. Molecular characterization of the tumors based on genetic mutations, on the identification of intracellular signaling pathways, and on their methylation profile is developing to better define their histological classification and give new insights into prognosis and treatment options.

## 2. Meningiomas

Meningiomas are extra-axial, slow-growing, and (usually) benign tumors. These tumors arise from meningothelial cells of the arachnoid layer, so they can be encountered anywhere this type of cell is localized. The most common locations (which make up about 50% of cases) are para-sagittal (28.8%), convexity (15.2%) and tuberculum sellae (12.8%). According to the statistical USA report by Ostrom et al. [1], meningiomas represent 36.6% of all primary CNS tumors and 53.2% of non-malignant primary CNS tumors. Although they have the reputation of benign lesions, meningiomas are often associated with a decreased quality of life (QoL) due to focal neurological deficits that may be related, and in 20% of cases, they display an aggressive behavior, even when the best standard of care is provided [3]. Meningiomas are most common in the elderly population (the median age at diagnosis is 66 years [1], and the occurrence risk increases with age) and have a female preponderance (1.8:1) [4]; however, grades II and III occur more often in males [5]. Only 1.5% of meningiomas occur in childhood or adolescence, and about 30% of these have an intra-ventricular localization [4]. Overall 5-year survival varies with age between 97% and 87.3% for non-malignant cases and between 85 and 50.2% for the malignant meningiomas [6]. Meningioma is considered a single type in the WHO CNS, and its broad morphological spectrum is reflected in 15 subtypes [2]. It is now emphasized that the criteria defining atypical or anaplastic (i.e., grade 2 and 3) meningioma should be applied regardless of the underlying subtype. As in prior classifications, chordoid and clear cell meningioma are noted to have a higher likelihood of recurrence than the average CNS WHO grade 1 meningioma and have hence been assigned to CNS WHO grade 2; however, larger and prospective studies would be helpful to validate these suggested CNS WHO grade 2 assignments and to suggest additional prognostic biomarkers. While papillary and rhabdoid features are often seen in combination with other aggressive features, more recent studies suggest that the grading of these tumors should not be carried out on the basis of rhabdoid cytology or papillary architecture alone. Several molecular biomarkers are associated with classification and grading of meningiomas, including SMARCE1 (clear cell subtype), BAP1 (rhabdoid and papillary subtypes), KLF4/TRAF (secretory subtype), TERT promoter mutation and/or homozygous deletion of CDKN2A/B (WHO grade 3), H3K27me3 loss of nuclear expression (potentially worse prognosis), and methylome profiling (prognostic subtyping) [7,8,9,10,11]. In meningiomas with a high risk of recurrence, we may include meningiomas with a high proliferation index, characterized by frequent mitotic figures and recurrence even after an apparently total removal, as well as particular subtypes of meningiomas such as atypical or rhabdoid meningiomas [4].

### 2.1. Risk Factors

The onset of meningiomas can be linked to environmental factors such as ionizing radiation. Ionizing radiation is the only established environmental risk factor, with higher risk in patients who have undergone radiation therapy [4]. It seems there is a genetic susceptibility to the development of radiation-induced meningiomas [12]. Numerous studies searching for the correlation between meningiomas and environmental factors (e.g., hormonal influence, diet, allergies, or phone use) have been conducted, but none with statistically significant results. Meningiomas occur with greater frequency in patients with germline mutations of genes such as type-2 neurofibromatosis (NF2) [13], type-1 neurofibromatosis (NF1, 19–24% of adolescent meningiomas) [4], type-1 multiple endocrine neoplasia (MEN1) [14], SMARCB1, LZTR1, or SMARCE1 genes [5].

### 2.2. Diagnosis

The gold standard for diagnosis and surveillance of meningiomas is magnetic resonance imaging (MRI). In MRI sequences, meningiomas appear as well-circumscribed and dural-based lesions, isointense to gray matter on non-contrast sequences, and homogeneously enhanced with gadolinium. Meningiomas may or not be associated with brain edema. A common but non-pathognomonic finding, especially in benign lesions, is the so-called dural tail: an enhancement of the dura due to a thickening of the dural layer adjacent to the mass. Moreover, MRI is very accurate in predicting venous sinus involvement (accuracy of about 90%) [15]. Meningiomas are usually single lesions. However, sometimes multiple lesions may suggest a genetic syndrome (e.g., NF2) or the presence of metastases. Other possible diseases to keep in consideration in differential diagnosis are [4,16]: pleomorphic xanthoastrocytoma which tend to be peripherally located and may have dural tail, dural metastases, gliosarcoma, Rosai-Dorfman disease, a connective tissue disorder with sinus histiocytosis, massive painless lymphadenopathy and dural based enhancing masses with characteristics similar to meningioma.

## 3. Treatment

### 3.1. Surgery

While radiological surveillance may be an acceptable strategy for many patients who present with asymptomatic incidental meningiomas [17], for growing and symptomatic tumors the standard of care remains maximal safe surgical resection. Surgical treatment allows for the removal of the lesion and, consequently, of the bulking effect; clinically, we can see the improvement of neurological functions and the resolution of seizures [18,19]. In meningioma surgery, the optimal goal is, whenever safely possible, the complete removal of the lesion and affected tissue. This minimizes neurological morbidity and allows for greater long-term control. Surgical treatment also makes it possible to obtain the correct histological diagnosis. Minimally invasive neurosurgery refers to the technological advances made in improving surgical access, thereby enabling neurosurgeons to reduce morbidity and greatly improve the precision of neurosurgical procedures. The introduction of the operating microscope and advances in neuroendoscopy have allowed for the refinement of lighting and the magnification of deep structures [20]. The use of micro-tools allows for the fine dissection of the neoplastic lesion from the nerves and vascular structures. Furthermore, advances in surgical neuronavigation and neuromonitoring allow, during the surgical approach, for minimum brain retraction, as well as the ability to highlight ischemic alterations and pressure variations in eloquent areas. In any case, surgical removal is limited by various factors, such as the location of the tumor and the incorporation within the lesion of nerves, venous sinuses, veins, and arteries. Simpson’s grading scale (Table 1) correlates to the extent of tumor resection, associated dural attachments, and any hyperostotic bone to local recurrence risk and five defined grades of resection, which are associated with distinct rates of recurrence [21].

In cases of partial occlusion of a venous sinus, the tumor component is not removed due to the high risk of hemorrhage, thrombosis, and gas embolism. Removal of meningiomas at the base of the skull, due to their relationship to bony anatomy (sphenoid wing, olfactory sulcus, saddle tubercle, ponto-cerebellar angle, or petroclival region) or those involving blood vessels and cranial nerves are at higher risk and require more advanced surgical techniques in order to minimize brain retraction and protect neurovascular structures [22]. Several midline anterior skull-base tumors are resected via an endoscopic endonasal approach. The advantages of this approach are the visualization of the ventral side of the deep skull-base tumor and safer resection which avoids traction of the brain tissue during the operation.

### 3.2. Radiation Therapy

Radiation therapy (RT) has commonly been utilized following subtotal resection surgery as an adjuvant therapy for recurrence of previously resected WHO grade II or grade III meningiomas [23]. Radiation therapy is individualized and must be chosen depending on meningioma size, proximity to critical structures, and any prior radiation to the same site. The goal of RT is to reduce meningioma’s proliferation and control its progress. Stereotactic radiotherapy (SRT) is defined as a method of external beam radiotherapy, in which a defined target volume is treated with a high radiation dose in up to 12 fractions, delivered on separate days of treatment. By means of SRT technology, it is possible to irradiate a specific target with a high dose in a single time; the dose of radiation absorbed outside the target area is inversely proportional to the distance from it, and the peripheral tissues around the focus are not damaged. Stereotactic radiosurgery (SRS) is the precise, single-session delivery of a therapeutically effective radiation dose to a certain target. SRS utilizes a single fraction of high-dose radiation but is associated with peculiar toxicities such as radiation-induced brain necrosis or intralesional hemorrhage [24]. Gamma knife radiosurgery (also known as stereotactic radiosurgery) focuses many tiny beams of radiation with extreme accuracy on a target. Each beam has very little effect on the brain tissue it passes through. Fractional stereotactic radiation therapy (FSRT) is administered over the course of several days, rather than in a single dose, reducing the dose exposure to normal brain tissue. SRS was developed by combining radiotherapy and stereotaxis. SRS is a widely accepted technique for small grade I or II lesions, while EBRT is recommended for grade III meningiomas, which require larger doses (50–60 Gy) to achieve local control [25,26]. Single-fraction SRS is typically used in meningiomas with a maximum diameter of <3 cm, located more than 3 mm from radiosensitive structures [27]. Although surgery remains the primary option, radiotherapy has become a first-line option for some meningiomas, particularly lesions of the cranial base that enclose vascular-nerve structures such as the optic nerve sheath or the cavernous sinus. Radiation therapy and fractional and hypofractionated stereotaxic radiosurgery, in single or multiple doses, have been shown to be beneficial for patients with a high rate of tumor control ranging from 85 to 100% at 5 years [28]. Side effects of stereotaxic radiotherapy for small tumors are mild [29,30], but cases of radionecrosis have been reported, and pituitary function should also be monitored after skull-base irradiation [31]. A long-term study of 290 consecutive patients showed progression-free control rates of 88.7 and 87.2% at 10 years and 20 years at follow-up, respectively, with adverse radiation effects in 3.1% of patients [32]. With the advent of SRS, surgical goals may be modified with plans for less-aggressive surgical resections, with the knowledge that radiosurgery offers an excellent option for long-term management of incompletely resected meningiomas. This aspect is particularly recommended for large meningiomas in surgically inaccessible locations. Such lesions may require pre-surgical planning for subtotal surgical resections, with planned adjuvant gamma knife radiosurgery to treat residual tumor due to its capacity to provide excellent long-term control of tumor growth while minimizing injury associated with tumor removal.

### 3.3. Chemotherapy

At long-term follow-up, up to 60% of meningiomas can recur after 15 years and exhibit aggressive behavior. RTOG 0539, a phase II trial that stratified meningioma risk based on pathologic grade and extent of resection, outlines options for postoperative management [33]. In RTOG 0539, meningiomas were stratified as follows: low risk—grade I and gross total resection (GTR) or subtotal resection (STR); intermediate risk—recurrent grade I or grade II after GTR; and high risk—STR or recurrent grade II and any grade III. Low-risk tumors demonstrated a progression-free survival (PFS) at 3 years of 92%, intermediate risk PFS of 94%, and high-risk PFS of 59% [34]. Due to their limited effectiveness, systemic therapies should be considered once all surgical and radiotherapy possibilities have been excluded and should be planned on an individual basis. There is little evidence in the literature to support systemic therapeutic treatment, and numerous clinical trials and case series have shown that chemotherapy has a minimal role and does not improve patients’ outcomes [35,36]. A major problem in interpreting the published literature on medical therapies for recurrent meningioma is the inclusion of different histology in reports of patients at various stages of their disease, ranging from newly diagnosed tumors to tumors that have relapsed after multiple surgeries and radiotherapy treatments and, in some cases, multiple chemotherapy regimens. Classic chemotherapeutic agents, including temozolomide, irinotecan, doxorubicin, ifosfamide, adriamycin, and vincristine, have not been shown to be effective against meningiomas [37,38]. Hydroxyurea, a ribonucleotide reductase inhibitor, can induce apoptosis and cell cycle arrest in the S-phase. An Italian randomized study showed the association between hydroxyurea, with or without imatinib, and recurrent or progressive meningiomas without, however, reaching conclusive data due to the small number of patients enrolled [39]. Hydroxyurea has shown stabilizing activity in only a few cases, but this has not been fully confirmed. It can be considered another adjuvant tool for atypical meningiomas if postoperative adjuvant radiotherapy cannot be applied. A recent study suggested that adjuvant treatment after STR of atypical meningiomas correlates with a longer PFS than conservative treatment and that there are no significant differences in PFS between hydroxyurea chemotherapy and radiotherapy after surgery [40]. In addition, hydroxyurea chemotherapy was shown to be effective in a retrospective study that analyzed 19 patients with atypical meningiomas. These were treated with hydroxyurea after GKR. The results of the present study suggest the safety and efficacy of HU after GKR with stabilization or shrinkage of atypical (grade II) meningiomas [41]. The possibility of systemic treatment as adjuvant therapy after surgery was also evaluated in a prospective study that enrolled 14 patients with malignant meningioma. After surgery and 2–4 weeks of radiotherapy (median dose 60 Gy), all patients were treated with cyclophosphamide, Adriamycin, and vincristine. This approach demonstrated moderate efficacy with partial response and disease stability in 3 and 11 patients, respectively, resulting in a median overall survival of 5.3 years and progression-free survival of 4.6 years [42].

## 4. Genetic and Biomarkers for Meningiomas

In the past years, thanks to new technologies in genomics, knowledge of genetic factors underlying the development of meningiomas has been significantly improved. With the identification of neurofibromin 2 (NF2) located on 22q12.2 (NF2 codes for the Merlin protein, which shows tumor suppressor properties) and the high percentage of patients (about 50–75%) with NF2 who develop one or more meningiomas, meningiomas represent one of the first types of tumors linked to a genomic driver [14].

Recently, thanks to the advent of next-generation sequencing (NGS), numerous additional somatic mutations have been identified (Table 2), and this has promising implications for new frontiers in target therapy. For this reason, to date, the mutational landscape can be divided into NF2-mutated (approximately 40–60% of cases) and non-NF2-mutated meningioma [43]. Among the non-NF2-mutated group, the most common oncogene in WHO grade I meningiomas is TNF receptor-activated factor 7 (TRAF7), located on chromosome 16p13 (mutated in nearly 25% of all meningiomas), followed by the mutation in codon K409Q of KLF4 (encountered in about 15% of benign meningiomas) [44,45]. The mutation of TRAF 7 is frequently associated with mutations of AKT1 (which encodes for a kinase that regulates cell proliferation) or KLF4, and this combination is often linked to grade 1 meningiomas [44,45]. AKT1 mutation occurs at the E17 location, triggering activation of the mTOR and ERK1/2 pathways. The AKT1 p.Glu 17 Lys mutation triggers the abnormal activation of the PI3K pathway, indicating a key role in meningiomas proliferation [46]. POLR2A mutant tumors show dysregulation of key meningeal identity genes, including WNT6 and ZIC1/ZIC4 [47]. Still, smoothened mutations (SMOs) trigger the Sonic Hedgehog signaling pathway promoting angiogenesis and tumor progression [48]. Mutations in Krüppel-like factor 4 (KLF4), a transcription factor in oncogenic activation, have been detected in about 50% of NF2-nonmutated meningiomas [45,49]. As shown in Table 1, in 20% of cases, the oncogenic mutations remain unclear. Notably, mutations of these genes in meningiomas occur to a large degree without concurrent alteration of NF2 or loss of chromosome 22 [50]. Other rarer mutations include SWI/SNF-related matrix-associated actin-dependent regulator of chromatin, subfamily B member 1 (SMARCB1), SMARCE1, and SUFU genes [51,52,53]. SMARCB1 mutation has also been shown to co-occur in NF2-mutated meningiomas [47]. In higher grade meningiomas, other genomic alterations with independent prognostic value have been reported, namely mutation of TERT promoter and deletions of CDKN2A/B [54,55]. The CDKN2A and BAP1 mutation seem to be associated with aggressive forms of meningioma [8,56], while the SMASRCE1 mutation is attributable to the onset of spinal meningiomas [52].

It has recently been hypothesized that the mutations and the direct molecular pathways activated in the processes of tumorigenesis could be related to their embryological cells of origin and, therefore, to the tumor site (Table 3) [44,47,57]. Okano et al. demonstrated that NF-2 mutant tumors originate from neural crest-derived arachnoid cells, while non-NF-2 tumors stem from the dorsal and paraxial mesoderm [58]. AKT1 mutations are demonstrable in anterior skull-base and convexity meningiomas, while SMO mutations are found in anterior skull-base meningiomas [44]. NF2 mutations are frequent in intraventricular meningiomas [59]. A large genetic analysis found high rates of NF-2 and POLR2A alterations in posterior fossa meningiomas [60].

In the few last decades, the genetic characteristics of meningiomas have become the target of investigation, and a large number of studies have focused on finding genes and biomarkers useful for predicting recurrence risk, malignant potential, and responsiveness to therapies. Tumor genomics has become a hot point in recent years because of the possibility of providing personalized and targeted therapies to patients. In 2016, the fourth edition of the WHO classification of CNS tumors [2] first used the molecular characteristics as parameters to diagnose this type of tumors, but only in 2021, with the fifth edition, did the WHO classification also introduce this parameter for meningiomas [2]. For example, secretory meningiomas (WHO grade 1) can be diagnosed not only for histological features, but also for the detection of the mutation of KLF4/TRAF 7 [2,23]. In clear cell meningioma, (WHO grade II) mutations of SMARCE1 occur frequently [7,61]. Moreover, according to the fifth edition, any meningioma with TERT promoter mutation and/or CDKN2A/B homozygous deletion is considered grade 3 [2,23]. In 2021, Nassiri et al. [62], by combining molecular characteristics, identified four groups of meningiomas: immunogenic type (MG1), benign NF2 wild-type (MG2), hypermetabolic type (MG3), proliferative type (MG4). The important aspect is that these four molecular groups seem to be able to predict clinical outcomes in a more accurate way (compared with already-existing classification schemes). Some key molecular alterations that could impact clinical management of patients with meningioma are [63] copy number alterations (CNAs), which determine alterations to the relations between oncogene and tumor suppressor activity, including: loss of chromosomes 22q, 1p, 14q, and 18q; mutations of genes NF2, TERTp, AKT1, PIK3CA, SMO, SUFU, TRAF7, KLF4, SMARCE1, BRCA1-associated protein 1 (BAP 1), Duchenne muscular dystrophy gene (DMD), PBRM1, POLR2A, and CDKN2A/B; homozygous deletions; epigenomic alterations; H3K27me3 alterations; TIMP3 methylation; and TP73 promoter (TP73p) methylation.

### Targeted Therapy

Currently, the most widely used pharmacological compounds in the treatment of high-grade and recurrent meningiomas or in cases in which surgical and/or radiotherapy treatment are found to be ineffective are anti-VEGF molecules and mTOR inhibitors. The expression of VEGF has been frequently associated with the grade of meningiomas and with their different histological types, but, to date, the information gathered to support this hypothesis appears to be conflicting. VEGF acts as a major vascular permeability factor and as a mitogen/survival promoter for endothelial cells and plays a key role in the formation of peritumoral edema [64]. Yamasaki et al. demonstrated that high levels of VEGF expression were significantly associated with tumor recurrence, suggesting that this factor is one of the main predictors of recurrence [65]. Sakuma et al. found evidence that VEGF expression was linked to development of peritumoral edema and to histological grade [66]. On the other hand, Barresi showed the absence of correlation between VEGF expression and WHO tumor grade [67]. Baxter et al. did not report any significant relationship, in 175 patients, between VEGF expression and histological meningioma grade [68]. In other studies, similarly, no significant correlation was found between VEGF expression and histological grade [69,70]. A recent study demonstrated that VEGF should not be used as a marker of severity or histological grade [71]. It is likely that expression of VEGF could be linked to the biomolecular and histologic characteristics of the various tumor subtypes. Since meningiomas are highly vascularized and express growth factors such as VEGF and PDGF, reduced angiogenesis may be useful for treatment.

It has been shown in in vitro meningioma extracts that these induce endothelial chemotaxis and the formation of small capillaries. In addition, VEGF plays an important role in the development of peritumor edema, which contributes to morbidity associated with high-grade meningiomas [72,73]. A phase II clinical trial in 2016 suggests that patients with refractory meningioma could benefit from a combined therapy with bevacizumab (a monoclonal antibody that inhibits binding between VEGF-A and VEGFR) and everolimus [74]. Antibodies to VEGF have been shown to be effective in controlling peritumoral edema and slowing lesion growth compared to other systemic therapies such as cytotoxic chemotherapy, somatostatin analogs, and tyrosine kinase inhibitors [75]. Unfortunately, studies using these targeted therapies alone or in combination in the treatment of meningioma have shown disappointing results [74,76,77]. In some studies using bevacizumab, a slight improvement in PFS was reported in patients with recurrent meningiomas [78,79]. Notably, a systematic review of bevacizumab in recurrent meningioma reports a median PFS of 15.3 months in recurrent atypical meningiomas and 3.7 months in anaplastic meningiomas. Franke et al. [80] suggested the use of bevacizumab only in specific circumstances such as treatment-refractory, high-grade cases, or cases with elevated vascularity; although surgery and radiosurgery represent the first therapeutic strategy for relapsing meningiomas, bevacizumab can delay relapse and thus can also serve as a potential neoadjuvant option to radiotherapy/surgery; patients with symptomatic multiple meningiomas are often difficult to treat due to the complexity of the surgery, the distance between lesions, and the predisposition to relapse. In these cases, resection is typically reserved for the largest symptomatic and surgically accessible lesions; therefore, adjuvant treatment such as radiotherapy/radiosurgery or chemotherapy (bevacizumab) may be potential options for patients with multiple meningiomas and radiation-induced meningiomas.

Sunitinib, unlike the previous single-target therapies, is a small-molecule kinase that inhibits different targets, such as VEGFR, PDGFR, FMS-like tyrosine kinase, KIT, CSF1R (macrophage colony-stimulating factor-1 receptor) and RET (a proto-oncogene), and most of them are highly represented in meningiomas. In 2015, a prospective and multicentric phase II trial [76] for recurrent and progressive atypical/anaplastic meningiomas was conducted. The authors concluded that sunitinib is active in recurrent atypical/malignant meningioma patients and suggested performing a randomized trial for a better evaluation of this molecule. In particular, an improvement in PFS of 42% at 6 months was observed compared to PFS of 5–30% at 6 months reported in the natural history. The limit is toxicity, with 60% of patients experiencing grade 3 toxicities, 32% of patients requiring dose reduction, and 22% of patients being removed from the study. Toxicities included CNS hemorrhage, GI symptoms, and anorexia [76].

mTOR is a protein kinase that forms the core of two proteins, mTORC1 and mTORC2. Both proteins possess regulatory activity on cell metabolism, growth and survival, and autophagic mechanisms [81]. The mTOR protein complex is an integral part of the PI3K complex, a pathway linked to the development of meningiomas. In an experimental study, the effects of the mTOR inhibitors sirolimus and tensirolimus were evaluated. Both inhibited the activity of mTORC1 and managed to decrease the viability and proliferation of neoplastic cells [82]. A recent clinical trial (CEVOREM) tested a combination of mTOR inhibitor everolimus and the somatostatin agonist octreotide on aggressive meningiomas. The results demonstrated an increase in PFS6 and a reduction in growth rate [83]. Interestingly, a reduction in the tumor growth rate (78% of patients) and an improvement in PFS at six months were observed. Other ongoing clinical trials (NCT03071874, NCT02831257) are evaluating the inhibitory activity of vistusertib against mTORC1 and MTORC2 [50]. Given their promising results, larger prospective phase III randomized clinical studies should be performed to draw conclusions about the role of this treatment in the context of meningiomas of different grades.

The expression of EGFR seems to be higher in low-grade meningiomas, but it is still unclear what the relation is between its expression and clinical prognosis [84]. The role of EGFR inhibitors in meningiomas is unclear. Some studies, however, have tried to evaluate the effects of EGFR inhibitor (e.g., gefitinib, erlotinib, or lapatinib [36,85]), but further randomized controlled trials should be carried out [63]. Erlotinib and gefitinib are small-molecule EGFR kinase inhibitors that have been investigated in phase II studies for recurrent meningioma. Although these treatments were well tolerated, neither gefitinib nor erlotinib appear to have significant activity in improving the PFS or overall survival of the patients analyzed [36]. Similarly, in a phase II study of imatinib, a small-molecule kinase inhibitor of the PDGF receptor, the imatinib was well tolerated but had no significant activity in recurrent meningiomas [86].

Evaluation of multi-targeted inhibitors and EGFR inhibitors in combination with other targeted molecular agents may be warranted. A phase II clinical trial of vatalanib (PTK787), a VEGFR and PDGFR inhibitor, was conducted by enrolling 25 patients with grade I, II, and III meningioma [77]. Each treatment cycle was 4 weeks with an MRI conducted every 8 weeks. On average, four cycles of PTK787 were administered to each patient. Vatalanib was safe, and minor toxicities including fatigue, hypertension and elevated transaminases were reported. Patients with atypical meningioma had progression-free survival (PFS) of 64.3%, median PFS of 6.5 months, and overall survival (OS) of 26 months; patients with malignant meningioma had PFS of 37.5%, median PFS of 3.6 months, and OS of 23 months [78]. It has been shown that NF2 mutant tumor cells interact with FAK inhibition. For this reason, a recent experimental study (NCT02523014) using GSK2256098 [87], a FAK inhibitor, in patients with NF2 mutant meningiomas, is now under evaluation [88].

## 5. Other Medical Treatments

### 5.1. Estrogen and Progesterone Receptor Antagonists

Meningiomas have a higher prevalence in females after puberty and during the reproductive years. Although there is no solid evidence to show a direct correlation between meningioma and reproductive hormone levels, the association between this type of tumor and reproductive hormones has been found in case reports and retrospective studies which are however limited by the small number of patients and confounding variables [89]. One study has shown a direct correlation between the number of pregnancies leading to childbirth and the risk of meningioma in women before age 50 [90,91]. In addition, a higher incidence of meningioma was found in breast cancer patients. Multiple meningiomas are less frequent but have a higher female predominance and a higher PR expression. While estrogen receptors (ERs) are expressed in 10% of meningiomas, PR is expressed in a higher percentage of meningiomas [92]. Higher-grade meningiomas tend to express more estrogen receptors, while benign meningiomas express progesterone receptor [93]. At present, it is not possible to make a definitive recommendation for the use of anti-estrogenic agents for meningioma due to lack of efficacy, as estrogen receptor inhibitors and anti-estrogen agents have not shown a strong effect [94]. A phase II study including 19 patients with unresectable and refractory meningiomas treated with tamoxifen, an estrogen receptor antagonist, showed no efficacy benefit in tumor growth control [95]. These findings are possibly due to the relatively infrequent estrogen receptor expression in meningioma. Some hormonal agents have been studied as possible systemic therapy in meningiomas. In 1991, a first study used mifepristone, a progesterone antagonist, for 2–31 months in 14 patients with unresectable meningioma. No high-grade drug toxicity was reported in any patient. Positive response of disease, defined as a reduction in tumor size on neuroimaging or an improvement in visual field examination, was documented in five patients. Three patients reported improvement in symptoms such as headache reduction or extraocular muscle function improvement [96]. It was reported that in three cases of meningioma treated with mifepristone, treatment was well tolerated, with significant and long-lasting stabilization or clinical (3/3) and radiological (2/3) response. All three patients remained stable after five to nine years of treatment [97]. A phase II clinical trial showed moderate improvement after mifepristone treatment in meningioma, especially in the male and premenopausal female subgroups of patients [96]. However, a larger randomized phase III trial for unresectable meningioma demonstrated the opposite result. Among 164 eligible patients, 80 were randomly assigned to mifepristone and 84 to placebo. There was no significant benefit between the groups in terms of failure-free or overall survival [98]. The failure of these chemotherapy agents in clinical studies is probably due to the wide molecular heterogeneity of meningiomas. The only subset showing a good response was the diffuse meningiomatosis group [97]. None of the studies evaluated the relationship between the PR isoform and mifepristone responsiveness.

### 5.2. Interferon-Alpha

Interferon-alpha is a biological agent that demonstrates a slight therapeutic benefit in recurrent meningiomas [99]. In the first in vitro meningioma experiments, it was reported that recombinant interferon-alpha showed inhibitory activity towards tumor cell growth, as it was able to inhibit DNA synthesis. Thirty-five patients with grade I recurrent meningiomas were enrolled in a study by Chamberlain and Glantz in 2008. All patients received daily INF-α subcutaneously. No treatment-related deaths or treatment delays were reported. Twenty-five patients (74%) had stable disease with a median tumor progression time of seven months, and nine patients (26%) had progressed. Median survival time was eight months. Although patients did not demonstrate a significant partial or complete radiographic response, IFNα had cytostatic activity resulting in significant palliation, as demonstrated by a 6-month PFS rate of 54% [100]. The outcomes of six patients with unresectable recurrent meningioma who received INF-α 2b for five days a week showed that one patient had minor tumor shrinkage, and four patients had stable disease that lasted up to 14 months. A longer and larger study of 12 patients with recurrent meningioma reported nine patients who had stable disease after treatment with INF-α that lasted up to eight years. Several studies have demonstrated stabilization of tumor growth, and a phase II study of recurrent meningiomas reported a slight improvement in PFS at 12 weeks with no improvement in overall survival rates.

### 5.3. Somatostatin Receptors

Overexpression of somatostatin receptors has been shown to be associated with aggressive tumors and higher relapse rates. Therefore, several somatostatin receptor inhibitors have been considered in the treatment of recurrent meningiomas, albeit with reduced therapeutic effects. In a study using a long-acting sandostatin inhibitor, the authors observed a partial radiographic response in 31% and a slight improvement in PFS6, with minimal toxicity [101], but other phase II clinical trials using sandostatin, octreotide, or other somatostatin receptor inhibitors have demonstrated minimal efficacy and not reported similar results [102]. Two ongoing trials (NCT03971461 and NCT04082520) are evaluating the antitumor efficacy of peptide receptor radionuclide therapy (PRRT) in refractory meningiomas. This pharmacological compound, already used in neuroendocrine tumors, targets SSTR2A, which is strongly expressed in meningiomas [103]. In any case, reduced radioactivity, cytopenia, and renal toxicity limit its therapeutic potential. Other hormone receptor inhibitors, including antiestrogen and antiprogesterone agents, have not demonstrated clinical benefits.

### 5.4. Immunotherapy

The increased understanding of the immune system and the tumor characteristics has led different authors to utilize immunotherapeutic modalities or combination therapy for the treatment of meningiomas [5]. The effects of IFN-α have been investigated because it has an important anti-angiogenetic activity that could be useful in tumors with high vascularization. Within the family of INF, IFN-α-2B is a leukocyte-produced molecule with known immunomodulatory and antiproliferative activities. A prospective clinical trial for grade I recurrent meningioma with IFN-α-2B [100] demonstrated that treatment with interferon-alpha for recurrent meningiomas is tolerated moderately well and is modestly effective. The discovery of PD1, a protein found on T cells that has a key role in the modulation of immune response, was essential in understanding the tumor development pathway. In fact, when PD-1 is bound to protein PD-L1, it contributes to keeping T cells from killing other cells (including cancer cells). Some anticancer drugs, called “immune checkpoint inhibitors”, are used to block PD-1 because when this protein is blocked, the ability of T cells to kill cancer cells is increased. It was found that in certain type of meningiomas (higher-grade), there are decreased levels of PD-1+ T-cells and an increased expression of PD-L1, which is associated with worse survival outcome [5]. Among these immune checkpoint inhibitors, we can observe pembrolizumab, a PD-1 inhibitor recently approved by FDA for the treatment of solid organ tumors in patients with an MMR deficiency (patients with a homozygous deletion of the DNA mismatch repair gene) and which can be used in patients with recurrent meningiomas and an MMR deficiency. A recent phase II trial demonstrated that pembrolizumab has promising efficacy on a subset of these tumors [104], but further studies are needed to identify the biological characteristics of these tumors that may drive response to immune-based therapies. It seems that there is an important response to immunotherapy in MMR-deficient patients, and it may be useful to screen this kind of patient to identify a subgroup that could obtain benefits from immunotherapy [105]. As said before, the PD-L1 is frequently over-expressed in high-grade meningiomas; for these reasons, even the use of PD-L1 inhibitors has been investigated. Among these, avelumab has induced antibody-dependent cellular cytotoxicity in in vitro studies using meningioma and NK-T cells [106]. Some studies involving this compound are yet ongoing.

### 5.5. MicroRNAs

Recent studies have demonstrated the potential role of microRNAs (miRNAs) in the biology of meningiomas, such that they can be included in future treatment strategies for meningiomas [107]. These short nucleotides may have antiangiogenic activity, tumor-suppressive activity, and suppression of immune evasion of tumors [108]. High expression of miR-190a and low expression of miR-29c-3p and miR-219-5p was correlated with higher recurrence rates [109]. Elevated levels of miR-335 increased cell growth and inhibited cell cycle arrest in the G0/G1 phase in vitro. It is probable that miR-335 plays an essential role in the proliferation of meningioma cells by directly targeting the retinoblastoma gene 1 (Rb1) signaling pathway [110]. MiR-21 is significantly upregulated in grade II and III meningiomas [111]. The expression of miR-224 is upregulated in grade III meningiomas, where it acts as an oncogene [112]. Overexpression of miR-34a-3p was shown to inhibit meningioma cell proliferation and induce cell apoptosis via decreased protein levels of SMAD4, FRAT1, and BCL2 in vitro [113]. However, the exact mechanisms of interactions between miRNAs and pathways are not yet fully understood, making their use complex.

## 6. Conclusions

To date, the treatment of meningiomas has mainly been focused on surgery and radiotherapy. The incidence, albeit reduced, of aggressive histotypes, or the presence of meningiomas located in critical areas of the brain or tenaciously adherent to vital structures, makes it necessary to adopt additional treatment methods to be used with those already existing. The recent advances in genetics and epigenetics and, in particular, the identification of specific genetic alterations have expanded our horizons to new classifications of these tumors and new therapeutic approaches. Molecular targeted therapies have certainly found particular interest, even if to date the results obtained are not very encouraging. Certainly, various growth vectors (PDGF, VEGF, EGF), their receptors, and their related pathways are implicated in these mechanisms (Ras/mitogen-activated protein kinase, phosphatidylinositol-3-kinase-Akt, phospholipase C-g1-protein kinase C pathways). In any case, to date, the specific significance of these biomarkers in tumor progression is still not fully understood. On the other hand, some mutations such as TRAF7, PI3KCA, AKT1, and SMOs are frequently observed in WHO grade 1 meningiomas and, consequently, their therapeutic potential appears somewhat limited. Research should be directed toward the identification of new tumor biomarkers and more specific pathways which, if appropriately targeted, could interfere with tumor growth. Currently, many clinical trials including targeted therapies and antiangiogenic agents are being investigated or under consideration, and the results of these studies could totally change the management and prognosis of these patients.

## Figures and Tables

**Table 1 cancers-15-04521-t001:** The Simpson Grade.

Simpson Grade	Description
Grade 1	Macroscopically complete tumor resection including removal of affected dura and underlying bone
Grade 2	Macroscopically complete tumor resection with coagulation of affected dura
Grade 3	Macroscopically complete tumor resection without removal of affected dura and underlying bone
Grade 4	Incomplete excision, subtotal tumor resection
Grade 5	Decompression with or without biopsy

**Table 2 cancers-15-04521-t002:** Most common mutations in WHO grade I meningiomas.

Mutation	Co-Occurring Mutations
NF2	SMARCB1
TRAF 7	AKT1 or KLF4 or PIK3CA
AKT1	/
POLR2A	/
SMARCB1	/
KLF4	/
SMO	/
PIK3CA	/
Unknown (20% of cases)	/

Mutations of TRAF 7 can overlap with AKT1, KLF4, and PIK3CA, but these last cannot overlap with each other; NF2: neurofibromin 2; TRAF 7: TNF receptor-associated factor 7 (a pro-apoptotic E3 ubiquitin ligase TNF receptor-associated factor 7); KLF4: Kruppel-like factor 4 (a pluripotency transcription factor); AKT1: v-Akt murine thymoma viral oncogene homolog 1 (a proto-oncogene); SMO: smoothened (a Hedgehog pathway signaling member); PIK3CA: oncogene.

**Table 3 cancers-15-04521-t003:** Biomarkers of meningiomas, tumor location, WHO grades, and targeted therapies.

Tumor Location	WHO Grade	Associated Mutations	Target Therapies
Convexity	I–III	22q, NF2, H3K27me3, SSTR2, BAP1, TERTp, CDKN2A/B, VEGFR	Sunitinib (22q), Everolimus-octreotide (SSTR2), Bevacizumab (VEGFR)
Anterior Skull Base	I–III	AKT1, PIK3CA, SMO, TRAF7	Everolimus-octreotide (SSTR2, AKT1, PI3K), Bevacizumab (VEGFR)
Central Skull Base	I	AKT1, PIK3CA, SMO, SUFU, TRAF7, H3K27me3, SSTR2, BAP1, TERTp, CDKN2A/B	Everolimus-octreotide (SSTR2, AKT1, PI3K), Bevacizumab (VEGFR)
Other Localizations	I–III	KLF-4, H3K27me3, SSTR2, TERTp, CDKN2A/B, BAP1, POLR2A	Everolimus-octreotide (SSTR2), Bevacizumab (VEGFR)
Spinal Meningiomas	I–III	22q, NF2, SMARCE1, H3K27me3, SSTR2, BAP1, TERTp, CDKN2A/B, VEGFR	Sunitinib (22q), Everolimus-octreotide (SSTR2), Bevacizumab (VEGFR)

NF2: Neurofibromin 2; H3K27me3: Trimethylation of lysine 27 on histone 3; SSTR2: Somatostatin receptor 2; BAP1: BRCA1-associated protein 1; TERTp: Telomerase reverse transcriptase promoter; CDKN2A/B: cyclin-dependent kinase inhibitor 2A/B; VEGFR: Vascular endothelial growth factor receptor; AKT1: Protein kinase B; PIK3CA: Phosphatidylinositol4,5-bisphosphate 3-kinase catalytic subunit alpha isoform; SMO: Smoothened; TRAF7: Tumor necrosis factor receptor associated factor 7; SUFU: Suppressor of fused homolog; KLF4: Krüppel-like factor 4; POLR2A: RNA polymerase II; TERT: Telomerase reverse transcriptase; SMARCE1: SWI/SNF Related, matrix associated, actin dependent regulator of chromatin, subfamily E, member 1.

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
