# Peer review of "Novel Advances in Treatment of Meningiomas: Prognostic and Therapeutic Implications"

_cancers, 2023, doi:10.3390/cancers15184521_

Round 1

Reviewer 1 Report

In the manuscript entitled Novel Advances in Treatment of Meningioma: Prognostic and Therapeutic Implications, the authors aim to review novel targeted  In the manuscript entitled Novel Advances in Treatment of Meningioma: Prognostic and Therapeutic Implications, the authors aim to review novel targeted treatments for non-surgical meningiomas.

Please consider the following minor revisions:

Ø  It would be interesting to summarize specific mutations in relations to specific meningioma subtypes and possible target therapy, maybe as a Table;

Ø  In the “Targeted Therapy” section, Authors underline that high expression of VEGFR is observed in high grade meningiomas, but they also report no correlation between VEGF and meningioma grade. It would be useful clarifying this apparent contradiction;

Ø  Being the toxicity the limiter to the use of everolimus, it would be better explaining if the association with octreotide helped reducing GI symptoms;

Ø  mTOR inhibitors other than everolimus have been tested, such as sirolimus and temsirolimus. Source: Pinker B, Barciszewska AM. mTOR Signaling and Potential Therapeutic Targeting in Meningioma. Int J Mol Sci. 2022 Feb 10;23(4):1978. doi: 10.3390/ijms23041978. PMID: 35216092; PMCID: PMC8876623;

Ø  To clinical purposes, Authors could include common treatment algorithms, including what molecules are usually employed first in meningioma treatment.

Approved with minor revision.

Ø  Please take care of some grammar mistakes in the text;

Author Response

We thank Reviewer for suggestions. We have carefully read reviewer’s comments, and have revised our manuscript accordingly.

Ø It would be interesting to summarize specific mutations in relations to specific meningioma subtypes and possible target therapy, maybe as a Table;

  • In accordance with your suggestion, we have inserted a new table (Table 3) in the revised manuscript.

Table 3. Biomarkers of meningiomas, tumor location, WHO grades, and targeted therapies.

Tumor

Location

WHO Grade

Associated Mutations

Target Therapies

Convexity

I-III

22q, NF2, H3K27me3, SSTR2, BAP1, TERTp, CDKN2A/B, VEGFR

Sunitinib (22q), Everolimus-octreotide (SSTR2), Bevacizumab (VEGFR)

Anterior Skull Base

I-III

AKT1, PIK3CA, SMO, TRAF7

Everolimus-octreotide (SSTR2, AKT1, PI3K), Bevacizumab (VEGFR)

Central Skull Base

I

AKT1, PIK3CA, SMO, SUFU, TRAF7, H3K27me3, SSTR2, BAP1, TERTp, CDKN2A/B

Everolimus-octreotide (SSTR2, AKT1, PI3K), Bevacizumab (VEGFR)

Others Localizations

I-III

KLF-4, H3K27me3, SSTR2, TERTp, CDKN2A/B, BAP1, POLR2A

Everolimus-octreotide (SSTR2), Bevacizumab (VEGFR)

Spinal meningiomas

I-III

22q, NF2, SMARCE1, H3K27me3, SSTR2, BAP1, TERTp, CDKN2A/B, VEGFR

Sunitinib (22q), Everolimus-octreotide (SSTR2), Bevacizumab (VEGFR)

NF2: Neurofibromin 2; H3K27me3: Trimethylation of lysine 27 on histone 3; SSTR2: Somatostatin receptor 2; BAP1: BRCA1 associated protein 1; TERTp: Telomerase reverse transcriptase promoter; CDKN2A/B: cyclin-dependent kinase inhibitor 2A/B; VEGFR: Vascular endothelial growth factor receptor; AKT1: Protein kinase B; PIK3CA: Phosphatidylinositol4,5-bisphosphate 3-kinase catalytic subunit alpha isoform; SMO: Smoothened; TRAF7: Tumor necrosis factor receptor associated factor 7; SUFU: Suppressor of fused homolog; KLF4: Krüppel-like factor 4;  POLR2A: RNA polymerase II; TERT: Telomerase reverse transcriptase; SMARCE1: SWI/SNF Related, matrix associated, actin dependent regulator of chromatin, subfamily E, member 1;

Ø In the “Targeted Therapy” section, Authors underline that high expression of VEGFR is observed in high grade meningiomas, but they also report no correlation between VEGF and meningioma grade. It would be useful clarifying this apparent contradiction;

  • We have noticed that data reported in the literature are not unique. Therefore, we have added further information on the subject in the corrected text:

Currently, the most widely used pharmacological compounds in the treatment of high-grade and recurrent meningiomas or, in the case in which surgical and/or radiotherapy treatment is found to be ineffective, are anti-VEGF molecules and mTOR inhibitors. The expression of VEGF has been frequently associated with the grade of meningiomas and with the different histological types, but, to date, the information gathered to support this hypothesis appears to be conflicting. The VEGF acts as a major vascular permeability factor, as a mitogen/survival promoter for endothelial cells, and plays a key role in the formation of peritumoral edema [65]. Yamasaki et al. demonstrated that high levels of VEGF expression were significantly associated with tumor recurrence, suggesting that this factor is one of the main predictors of recurrence [66]. Sakuma et al. evidenced that VEGF expression was linked to development of peritumoral edema and to histological grade [67]. On the other hand, Barresi showed the absence of correlation between VEGF expression and WHO tumor grade [68]. Baxter et al. did not show any significant relationship, in 175 patients, between VEGF expression and histological meningioma grade [69]. In other studies, similarly, no significant correlation was found between VEGF expression and histological grade [70-71]. A recent study, evidenced that VEGF should not be used as a marker of severity or histological grade [72]. Probably, expression of VEGF could be linked to the biomolecular and histologic characteristics of the various tumor subtypes.

  • 65. Caffo, M.; Pino, M.A.; Caruso, G.; Tomasello, F. Antisense Molecular Therapy in Cerebral Gliomas. Analyt. Oncol. 2012, 1:135-144.
  • Yamasaki, F.; Yoshioka, H.; Hama, S.; Sugiyama, K.; Arita, K.; Kurisu, K. Recurrence of Meningiomas. Cancer 2000, 89, 1102-1110.
  • Sakuma, T.; Nakagawa, T.; Ido, K.; Takeuchi, H.; Sato, K.; Kubota, T. Expression of Vascular Endothelial Growth Factor-A and mRNA Stability Factor HuR in Human Meningiomas. J. Neurooncol. 2008, 88, 143-155.
  • 68. Barresi, V.; Tuccari, G. Increased Ratio of Vascular Endothelial Growth Factor to Semaphorin3A is a Negative Prognostic Factor in Human Meningiomas. Neuropathology 2010, 30, 537-546. doi: 10.1111/j.1440-1789.2010.01105.x.
  • 69. Baxter, D,S.; Orrego, A.; Rosenfeld, J.V.; Mathiesen, T. An Audit of Immunohistochemical Marker Patterns in Meningioma. Clin. Neurosci. 2014, 21, 421-426.
  • 70. Pistolesi, S.; Fontanini, G.; Camacci, T.; de Ieso, K.; Boldrini, L.; Lupi, G.; et al. Meningioma-Associated Brain Oedema: The Role of Angiogenic Factors and Pial Blood Supply. Neurooncol. 2002, 60, 159-164.
  • Denizot, Y.; de Armas, R.; Caire, F.; Moreau, J.J.; Pommepuy, I.; Truffinet, V.; et al. The Quantitative Analysis of bFGF and VEGF by ELISA in Human Meningiomas. Mediators Inflamm. 2006, 2006, 36376.
  • 72. Winter, R.C.; Antunes, A.C.M.; de Oliveira, F.H. The Relationship Between Vascular Endothelial Growth Factor and Histological Grade in Intracranial Meningioma. Neurol. Int. 2020, 11.

Ø  Being the toxicity the limiter to the use of everolimus, it would be better explaining if the association with octreotide helped reducing GI symptoms;

  • From a careful literature search, we have not found data to support the hypothesis that octreotide, associated with everolimus, can reduce GI symptoms.

Ø  mTOR inhibitors other than everolimus have been tested, such as sirolimus and temsirolimus. Source: Pinker B, Barciszewska AM. mTOR Signaling and Potential Therapeutic Targeting in Meningioma. Int J Mol Sci. 2022 Feb 10;23(4):1978. doi: 10.3390/ijms23041978. PMID: 35216092; PMCID: PMC8876623;

  • In the revised manuscript, we have added additional data on mTOR inhibitors (sirolimus, temsirolimus). We have also added the reference you suggested.

mTOR is a protein kinase that forms the core of two proteins, mTORC1 and mTORC2. Both proteins possess regulatory activity on cell metabolism, growth and survival and on autophagic mechanisms [82]. The mTOR protein complex is an integral part of the PI3K complex, a pathway linked to the development of meningiomas. In an experimental study, the effects of the mTOR inhibitors sirolimus and tensirolimus were evaluated. Both inhibiting the activity of mTORC1 managed to decrease the viability and proliferation of neoplastic cells [83]. A recent clinical trial (CEVOREM), used in aggressive meningiomas, a combination of mTOR inhibitor everolimus and the somatostatin agonist octreotide. The results demonstrated an increase in PFS6 and a reduction in growth rate [84]. Interestingly, the reduction in the tumor growth rate (78% of patients) and the improvement in PFS at six months were observed. Other ongoing clinical trials (NCT03071874, NCT02831257) are evaluating the inhibitory activity of vistusertib against mTORC1 and MTORC2 [50]. Given the promising results, larger prospective phase III randomized clinical studies should be performed to draw conclusions about the role of this treatment in the context of meningiomas from different grades.

  • 82. Pinker, B.; Barciszewska, A.M. mTOR Signaling and Potential Therapeutic Targeting in Meningioma. J. Mol. Sci. 2022, 23, 1978. doi: 10.3390/ijms23041978.
  • 83. Pachow, D.; Andrae, N.; Kliese, N.; Angenstein, F.;  Stork, O.; Wilisch-Neumann, A.; et al. mTORC1 Inhibitors Suppress Meningioma Growth in Mouse Models. Cancer Res. 2013, 19, 1180-1189. doi: 10.1158/1078-0432.CCR-12-1904. 
  • 84. Graillon, T.; Sanson, M.; Campello, C.; Idbaih, A.; Peyre, M.; Peyrière, H.; et al. Everolimus and Octreotide for Patients with Recurrent Meningioma: Results from the Phase II CEVOREM Trial. Clin. Cancer Res. Off. J. Am. Assoc. Cancer Res. 2020, 26, 552–557, doi:10.1158/1078-0432.CCR-19-2109.
  • 50. Bi, W.L.; Zhang, M.; Wu, W.W.; Mei, Y.; Dunn, I.F. Meningioma Genomics: Diagnostic, Prognostic, and Therapeutic Applications. Front. Surg. 2016, 3, 40, doi:10.3389/fsurg.2016.00040.

Ø To clinical purposes, Authors could include common treatment algorithms, including what molecules are usually employed first in meningioma treatment.

  • From a search of the relevant literature, the pharmacological compounds most used in the treatment of meningiomas (relapses, unsuitable for surgical and/or radiotherapy) seem to be mTOR and anti-VEGF inhibitors. With the data present in the literature today, we think it is still premature to hypothesize guidelines on the use of these new therapeutic protocols (to confirm the actual results, to evaluate any toxic effects, to test new drugs).

Reviewer 2 Report

Simple Summary: I’m not sure I would agree that “Meningioma's molecular biology, genetics, and epigenetics are still poorly understood.”, or later, “very little is understood about the crucial molecular alterations favoring meningioma growth and their molecular pathogenesis.” You can always argue that a better understanding would be helpful in advancing treatments, but at this point there have been quite a few studies on meningioma epigenetics and genomics, and most of the prognostic alterations (TERT, CDKN2A, loss of chr22q/1p, DNA methylation subgroups) are well characterized.

The manuscript needs to be carefully edited for grammar/syntax. Some examples from the first page: “Although the status of benign lesions, meningiomas are…”, “slow-growing and (usually) benign tumor.” (should be plural), “and about 30% of these has an intra-ventricular” (should be ‘have’), “promoter mutation61 and/or homozygous” (should not contain ’61’), “CDKN2A/B62” (should not contain 62), “"risk factor, with higher risk in patient undergone radiation therapy in childhood" (should be ‘patients’).

It is not clear what this means:  “In the light of these observations, the identification  of specific proteins can be considered a valid and promising treatment strategy”

In “Several molecular biomarkers are associated with classification and grading of meningiomas…”, only a single citation is provided, and it appears to be only relevant to SMARCE1.

“xanthoastrocitoma” is spelled incorrectly

“remains maximal safe surgical resection (RFS).” What does ‘RFS’ stand for?

"greatly improve the accuracy of neurosurgical procedures." I don’t think minimally invasive surgery would improve accuracy. ‘Precision’ might be a better word.

“The goal in surgical management of meningiomas is complete resection of the tumor and affected tissues, both hard and bone” would qualify this statement by saying as ‘safely feasible’.

It is not clear how you define “Stereotaxic radiotherapy (SRT)” vs. “single-fraction stereotactic radiation (SRS)”.  The latter is also abbreviated twice in the manuscript (lines 160 and 171). Additionally, a better definition should be given for “Fractional stereotaxis radiation therapy (FSRT)”. It would be best to describe what ‘fractional’ means in the context of radiotherapy, which will make the differences between these therapies more clear to the reader.

“SRS is a widely accepted technique for small grade I or II lesions, while EBRT is recommended for grade III meningiomas, which require larger doses (50–60 Gy) to achieve local control.” A citation should be provided for this.

Line 191: ‘Gamma Knife’ should be introduced as a type of SRS.

Table 1: POLR2A and SMARCB1 should be added, as they have similar prevalence as SMO, KLF4 and PIK3CA.

“Recently, thanks to the advent of next-generation sequencing (NGS), numerous additional somatic mutations have been identified (Table 1) and this has promising implications for the new frontiers in target therapy.” References should be provided to the original studies for the genes in Table 1. Later in the paragraph there are several references to review articles.

“Mutations in Krüppel-like factor 4 (KLF4), a transcription factor in oncogenic activation have been detected in about 50% of NF2-nonmutated meningiomas.”  This percentage is way too high. What is the source of this?

“while the BAP-1 mutation seems to be frequent in spinal meningiomas [41].” This statement seems incorrect. I am not able to find a comment on this in the provided reference.

“Mutations of the SMO and SUFU genes are observed in meningiomas of the skull base, while the TRAF7 mutation is found in meningiomas of the anterior and medial skull base [42].” It is not clear how this relates mutation to the site of origin? Also, again a review article is cited here. It would be much better to cite primary research studies than secondary reviews. Similar statement for lines 294-302.

"In grade 2 meningiomas is frequent the mutation of SMARCE1 (97%)" This appears to state that 97% of grade 2 meningiomas have SMARCE1 mutations, which is not accurate.

Lines 324 – 330: there are several places where semi-colons are used instead of periods.

For the section “4.1. Targeted Therapy” it might improve readability of the single large block of text was broken into paragraphs for each of the drugs discussed.

“Interferon-alpha is a biological agent that demonstrates a slight therapeutic benefit in recurrent meningiomas.” There should be a reference for this.

“slight improvement in PFS6, with” Does this refer to PFS at 6 months? This should be defined.

“little research has been done to identify new and alternative treatment modalities". This statement is not accurate. You could say that no other therapies besides radiation/surgery have been found to be effective with conclusive evidence.

“However, very little is known about the pathogenesis of meningiomas and the molecular alterations promoting meningioma proliferation.” As discussed above, this is not accurate.

The discussion of micro-RNAs should be moved to the main text (not the conclusions)

Only minor issues in understanding the text, however there are many areas where the syntax/grammar could be improved.

Author Response

We thank Reviewer for your suggestions. We have carefully read reviewer’s comments, and have revised our manuscript accordingly.

  • Simple Summary: I’m not sure I would agree that “Meningioma's molecular biology, genetics, and epigenetics are still poorly understood.”, or later, “very little is understood about the crucial molecular alterations favoring meningioma growth and their molecular pathogenesis.” You can always argue that a better understanding would be helpful in advancing treatments, but at this point there have been quite a few studies on meningioma epigenetics and genomics, and most of the prognostic alterations (TERT, CDKN2A, loss of chr22q/1p, DNA methylation subgroups) are well characterized.
  • In accordance with the reviewer's suggestion, we have removed the two sentences.

  • The manuscript needs to be carefully edited for grammar/syntax. Some examples from the first page: “Although the status of benign lesions, meningiomas are…”, “slow-growing and (usually) benign tumor.” (should be plural), “and about 30% of these has an intra-ventricular” (should be ‘have’), “promoter mutation61 and/or homozygous” (should not contain ’61’), “CDKN2A/B62” (should not contain 62), “"risk factor, with higher risk in patient undergone radiation therapy in childhood" (should be ‘patients’).
  • The manuscript, as advised by the reviewer, has undergone a careful review of the English language and syntax/grammar.

  • It is not clear what this means: “In the light of these observations, the identification of specific proteins can be considered a valid and promising treatment strategy”
  • This sentence has been removed from the text.

  • In “Several molecular biomarkers are associated with classification and grading of meningiomas…”, only a single citation is provided, and it appears to be only relevant to SMARCE1.
  • In the revised text, we have added other pertinent references:
  • Shankar, G.M.; Abedalthagafi, M.; Vaubel, R.A.; Merrill, P.H.; Nayyar, N.; Gill, C.M.; et al. Germline and Somatic BAP1 Mutations in High-Grade Rhabdoid Meningiomas. Neuro-Oncology2017, 19, 535–545. doi: 10.1093/neuonc/now235.
  • 9.Williams, E.A., Wakimoto, H., Shankar, G.M. et al.Frequent inactivating mutations of the PBAF complex gene PBRM1 in meningioma with papillary features. Acta Neuropathol. 2020, 140, 89–93. org/10.1007/s00401-020-02161-7.
  • 10. Lee, Y.S.; Lee, Y.S. Molecular Characteristics of Meningiomas. Pathol. Transl. Med.2020, 54, 45–63. doi: 10.4132/jptm.2019.11.05.
  • 11. Nassiri, F.; Wang, J.Z.; Singh, O.; Karimi, S.; Dalcourt, T.; Ijad, N.; et al. Loss of H3K27me3 in Meningiomas. Neuro Oncol.2021, 23, 1282–1291. doi: 1093/neuonc/noab036.

  • “xanthoastrocitoma” is spelled incorrectly
  • The term xanthoastrocytoma has been corrected.

  • “remains maximal safe surgical resection (RFS).” What does ‘RFS’ stand for?
  • The acronym RFS has been removed.

  • "greatly improve the accuracy of neurosurgical procedures." I don’t think minimally invasive surgery would improve accuracy. ‘Precision’ might be a better word.
  • As you suggested, we have replaced the word "accuracy" with "precision".

  • “The goal in surgical management of meningiomas is complete resection of the tumor and affected tissues, both hard and bone” would qualify this statement by saying as ‘safely feasible’.
  • In consideration of your suggestion we have thus modified the sentence: "In meningioma surgery, the optimal goal is, whenever and safely possible, the complete removal of the lesion and affected tissue”.

  • It is not clear how you define “Stereotaxic radiotherapy (SRT)” vs. “single-fraction stereotactic radiation (SRS)”.  The latter is also abbreviated twice in the manuscript (lines 160 and 171). Additionally, a better definition should be given for “Fractional stereotaxis radiation therapy (FSRT)”. It would be best to describe what ‘fractional’ means in the context of radiotherapy, which will make the differences between these therapies moreclear to the reader.
  • In accordance with your comments and suggestions, we have revised section 3.2 "Radiation Therapy":

Radiation therapy (RT) has commonly been utilized as an adjuvant therapy following subtotal resection surgery, as treatment of recurrence for previously resected meningiomas, for WHO grade II or grade III meningioma [23]. Radiation therapy is individualized and must be chosen depending of meningioma size, proximity to critical structures, and any prior radiation to the same site. The goal of RT is to reduce meningioma’s proliferation and control its progress. Stereotactic radiotherapy (SRT) is defined as a method of external beam radiotherapy, in which a defined target volume is treated with a high radiation dose in up to 12 fractions delivered on separate days of treatment. By means of SRT technology, it is possible to irradiate a specific target with a high dose in a single time; the dose of radiation absorbed outside the target area is inversely proportional to the distance from it and the peripheral tissues around the focus are not damaged. Stereotactic radiosurgery (SRS) is the single-session, precise delivery of a therapeutically effective radiation dose to a certain target. SRS utilizes a single fraction of high dose radiation but is associated with peculiar toxicities such as radiation-induced brain necrosis or intralesional hemorrhage [24]. Gamma Knife radiosurgery (also known as stereotactic radiosurgery) focuses many tiny beams of radiation on a target with extreme accuracy. Each beam has very little effect on the brain tissue it passes through. Fractional stereotactic radiation therapy (FSRT) is administered over the course of several days, rather than in a single dose, reducing the exposure dose of normal brain tissue. SRS was developed by combining radiotherapy and stereotaxis. SRS is a widely accepted technique for small grade I or II lesions, while EBRT is recommended for grade III meningiomas, which require larger doses (50–60 Gy) to achieve local control [25-26]. Single fraction SRS is typically used in meningiomas with a maximum diameter of <3 cm and located more than 3 mm from radiosensitive structures [27]. Although surgery remains the primary option, radiotherapy has become a first-line option for some meningiomas, particularly lesions of the base cranial that enclose vascular-nerve structures such as the optic nerve sheath or the cavernous sinus. Radiation therapy and fractional and hypofractionated stereotaxic radiosurgery, in single or multiple doses, have been shown to be beneficial for patients with a high rate of tumor control ranging from 85 to 100% at 5 years [28]. Side effects of stereotaxic radiotherapy for small tumors are mild [29-30], but cases of radionecrosis have been reported and pituitary function should also be monitored after skull base irradiation [31]. A long-term study of 290 consecutive patients showed progression-free control rates of 88.7 and 87.2% at 10 years and 20 years at follow up, respectively, with adverse radiation effects in 3.1% of patients [32]. With the advent of SRS, surgical goals may be modified with plans for less aggressive surgical resections with the knowledge that radiosurgery offers on excellent option for long-term management of incompletely resected meningiomas. This aspect is particularly characterized for large meningiomas in surgically inaccessible locations. Such lesions may require pre-surgical planning for subtotal surgical resections, with planned adjuvant Gamma Knife radiosurgery to treat residual tumor, knowing its capability to provide excellent long-term control of tumor growth while minimizing injury associated with tumor removal.

  • 24. Jimenez, R.B.; Alexander, B.M.; Mahadevan, A.; Niemierko, A.; Rajakesari, S.; Arvold, N.D.; et al. The impact of Different Stereotactic Radiation Therapy Regimens for Brain Metastases on Local Control and Toxicity.  Rad. Oncol. 2017, 2, 391–397. doi: 10.1016/j.adro.2017.05.008.
  • 25. Cohen-Inbar, O.; Lee, C.C.; Sheehan, J.P. The Contemporary Role of Stereotactic Radiosurgery in the Treatment of Meningiomas. Neurosurg. Clin. N. Am. 2016, 27, 215-228.
  • 26. Buerki, R.A.; Horbinski, C.M.;Kruser, T.; Horowitz, P.M.; James, C.D.; Lukas, R.M. An Overview of Meningiomas. Future Oncol2018, 14, 2161–2177. doi: 2217/fon-2018-0006.

  • “SRS is a widely accepted technique for small grade I or II lesions, while EBRT is recommended for grade III meningiomas, which require larger doses (50–60 Gy) to achieve local control.” A citation should be provided for this.
  • With regard to the definition " SRS is a widely accepted technique for small grade I or II lesions, while EBRT is recommended for grade III meningiomas, which require larger doses (50–60 Gy) to achieve local control [25-26]." We have introduced the related references in the revised text:
  • 25. Cohen-Inbar, O.; Lee, C.C.; Sheehan, J.P. The Contemporary Role of Stereotactic Radiosurgery in the Treatment of Meningiomas. Neurosurg. Clin. N. Am. 2016, 27, 215-228.
  • 26. Buerki, R.A.; Horbinski, C.M.;Kruser, T.; Horowitz, P.M.; James, C.D.; Lukas, R.M. An Overview of Meningiomas. Future Oncol. 2018, 14, 2161–2177. doi: 2217/fon-2018-0006.

  • Line 191: ‘Gamma Knife’ should be introduced as a type of SRS.
  • According to your suggestion we have introduced the "Gamma Knife" as a type of SRS.

  • Table 1: POLR2A and SMARCB1 should be added, as they have similar prevalence as SMO, KLF4 and PIK3CA.
  • We have added in table 1 POLR2A and SMARCB1

  • “Recently, thanks to the advent of next-generation sequencing (NGS), numerous additional somatic mutations have been identified (Table 2) and this has promising implications for the new frontiers in target therapy.” References should be provided to the original studies for the genes in Table 2. Later in the paragraph there are several references to review articles.
  • In the revised text we have introduced the specific references:

Recently, thanks to the advent of next-generation sequencing (NGS), numerous additional somatic mutations have been identified (Tab. 2) and this has promising implications for the new frontiers in target therapy. For this reason, to date, the mutational landscape can be divided into NF2-mutated (approximately 40-60% of cases) and non-NF2-mutated meningioma [43]. Among the non-NF2-mutated group, the most common oncogene in WHO grade I meningiomas is TNF receptor-activated factor 7 (TRAF7), located on chromosome 16p13 (mutated in nearly 25% of all meningiomas), followed by the mutation in codon K409Q of KLF4 (encountered in about 15% of benign meningiomas) [44-45]. The mutation of TRAF 7 is frequently associated with mutations of AKT1 (which encodes for a kinase that regulates cell proliferation) or KLF4, and this combination is often linked with grade 1 meningiomas [44-45]. AKT1 mutation occurs at the E17 location triggering activation of the mTOR and ERK1/2 pathways. The AKT1 p.Glu 17 Lys mutation triggers the abnormal activation of the PI3K pathway showing a key role in meningiomas proliferation [46]. POLR2A mutant tumors show dysregulation of key meningeal identity genes, including WNT6 and ZIC1/ZIC4 [47]. Still, Smoothened (SMO) mutations triggers the Sonic hedgehog signaling pathway promoting angiogenesis, and tumor progression [48]. Mutations in Krüppel-like factor 4 (KLF4), a transcription factor in oncogenic activation have been detected in about 50% of NF2-nonmutated meningiomas [45,49]. As shown in Tab 1, in 20% of cases, the oncogenic mutations remain unclear. Notably, mutations of these genes in meningiomas occur to large degree without concurrent alteration of NF2 or loss of chromosome 22 [36]. Other rarer mutations include SWI/SNF related matrix associated, actin dependent regulator of chromatin, subfamily B member 1 (SMARCB1), SMARCE1, and SUFU genes [51-53]. In higher grade meningiomas, are reported other genomic alterations with independent prognostic value: mutation of TERT promoter and deletions of CDKN2A/B [54-55]. The CDKN2A and BAP1 mutation seem to be associated with aggressive forms of meningioma [8,56], while the SMASRCE1 mutation is attributable to the onset of spinal meningiomas [52].

  • 43. Dumanski, J.P.; Rouleau, G.A.; Nordenskjold, M.; Collins, V.P. Molecular Genetic Analysis of Chromosome 22 in 81 Cases of Meningioma. Cancer Res. 1990, 50, 5863–5867.
  • 44. Clark, V.E.;Erson-Omay, E.Z.; Serin, A.; Yin, J.; Cotney, J.; Ozduman, K.; et al. Genomic Analysis of non-NF2 Meningiomas Reveals Mutations in TRAF7, KLF4, AKT1, and SMO. Science 2013, 339, 1077-1080. doi: 10.1126/science.1233009.
  • 45. Reuss, D.E.;Piro, R.M.; Jones, D.T.; Simon, M.; Ketter, R.; Kool, M.; et al. Secretory Meningiomas are Defined by Combined KLF4 K409Q and TRAF7 Mutations. Acta Neuropathol. 2013, 125, 351-358. doi: 10.1007/s00401-013-1093-x.
  • 46. Pachow, D.; Wick, W.; Gutmann, D.H.; Mawrin, C. The mTOR Signaling Pathway as a Treatment Target for Intracranial Neoplasms. Neuro-Oncology 2015, 17, 189–199.
  • 47. Clark, V.E.; Harmanc, A.S.;, Bai. H.; et al. Recurrent Somatic Mutations in POLR2A Define a Distinct Subset of Meningiomas. Nat Genet. 2016, 48, 1253–1259.
  • 48. Brastianos, P.K.; Horowitz, P.M.; Santagata, S. Jones, R.T.; McKenna, A.; Getz, G.; et al. Genomic Sequencing

of Meningiomas Identifies Oncogenic SMO and AKT1 Mutations. Nat. Genet. 2013, 45, 285–289. doi: 10.1038/ng.2526.

  • 49. Moussalem, C.; Massaad, E.; Minassian, G.B.; Ftouni. L.; Bsat, S.; El Houshiemy, M.N.; et al. Meningioma Genomics: a Therapeutic Challenge for Clinicians. Integr. Neurosci. 2021, 20, 463-469.  doi: 10.31083/j.jin2002049.
  • 50. Bi, W.L.; Zhang, M.; Wu, W.W.; Mei, Y.; Dunn, I.F. Meningioma Genomics: Diagnostic, Prognostic, and Therapeutic Applications. Front. Surg. 2016, 3, 40, doi:10.3389/fsurg.2016.00040.
  • Wang, S.A.; Jamshidi, A.O.; Oh, N.; Sahyouni, R.; Nowroozizadeh, B.; Kim. R.; et al. Somatic SMARCB1 Mutation in Sporadic Multiple Meningiomas: Case Report. Front. Neurol. 2018, 9, 919. doi: 10.3389/fneur.2018.00919. 
  • 52. Smith, M.J.; O’Sullivan, J.; Bhaskar, S.S.; Hadfield, K.D.; Poke, G.; Caird, J.; et al. Loss-of-Function Mutations in SMARCE1 Cause an Inherited Disorder of Multiple Spinal Meningiomas.  Genet.201345, 295–298.
  • Aavikko, M.; Li, S.;  Saarinen, S.; Alhopuro, P.; Kaasinen, E.; Morgunova, E.; et al.  Loss of SUFU Function in Familial Multiple Meningioma. Am. J. Hum. Genet. 2012, 91, 520-526. doi: 10.1016/j.ajhg.2012.07.015.
  • 54. Goutagny, S.; Nault, J.C.; Mallet, M.; Henin, D.; Rossi, J.Z.; Kalamarides, M. High Incidence of Activating TERTPromoter Mutations in Meningiomas Undergoing Malignant Progression. Brain Pathol. 2014, 24, 184-9. doi: 10.1111/bpa.12110.
  • 55. Barresi, V.; Simbolo, M.; Fioravanzo, A.; Piredda, M.L.; Caffo, M.; Ghimenton, C.; et al. Molecular Profiling of 22 Primary Atypical Meningiomas Shows the Prognostic Significance of 18q Heterozygous Loss and CDKN2A/B Homozygous Deletion on Recurrence-Free Survival. Cancers 2021, 13, 903, doi:10.3390/cancers13040903.
  • 56. Wang, J.Z.; Patil, V.; Liu, J.; Dogan, H.; Tabatabai, G.; Yefet, L.S.; et al. Increased mRNA Expression of CDKN2A is a Transcriptomic Marker of Clinically Aggressive Meningiomas. Acta Neuropathol. 2023, 46, 145-162. doi: 10.1007/s00401-023-02571-3. 
  • 8. Shankar, G.M.; Abedalthagafi, M.; Vaubel, R.A.; Merrill, P.H.; Nayyar, N.; Gill, C.M.; et al. Germline and Somatic BAP1 Mutations in High-Grade Rhabdoid Meningiomas. Neuro-Oncol. 2017, 19, 535–545, doi:10.1093/neuonc/now235.

  • “Mutations in Krüppel-like factor 4 (KLF4), a transcription factor in oncogenic activation have been detected in about 50% of NF2-nonmutated meningiomas.”  This percentage is way too high. What is the source of this?
  • In relation to the phrase "Mutations in Krüppel-like factor 4 (KLF4), a transcription factor in oncogenic activation have been detected in about 50% of NF2-nonmutated meningiomas [45,49]." we have inserted the specific references: 
  • 45. Reuss, D.E.;Piro, R.M.; Jones, D.T.; Simon, M.; Ketter, R.; Kool, M.; et al. Secretory Meningiomas are Defined by Combined KLF4 K409Q and TRAF7 Mutations. Acta Neuropathol. 2013, 125, 351-358. doi: 10.1007/s00401-013-1093-x.
  • 49. Moussalem, C.; Massaad, E.; Minassian, G.B.; Ftouni. L.; Bsat, S.; El Houshiemy, M.N.; et al. Meningioma Genomics: a Therapeutic Challenge for Clinicians. Integr. Neurosci. 2021, 20, 463-469.  doi: 10.31083/j.jin2002049.

  • “while the BAP-1 mutation seems to be frequent in spinal meningiomas [41].” This statement seems incorrect. I am not able to find a comment on this in the provided reference.
  • We thank the reviewer for his right observation. Therefore, we have changed the phrase " The CDKN2A mutation has recently been identified in the most aggressive meningiomas, while the BAP-1 mutation seems to be frequent in spinal meningiomas [41]." to "The CDKN2A and BAP1 mutation seem to be associated with aggressive forms of meningioma [8,56], while the SMASRCE1 mutation is attributable to the onset of spinal meningiomas [52].
  • 56. Wang, J.Z.; Patil, V.; Liu, J.; Dogan, H.; Tabatabai, G.; Yefet, L.S.; et al. Increased mRNA Expression of CDKN2A is a Transcriptomic Marker of Clinically Aggressive Meningiomas. Acta Neuropathol. 2023, 46, 145-162. doi: 10.1007/s00401-023-02571-3. 
  • 8. Shankar, G.M.; Abedalthagafi, M.; Vaubel, R.A.; Merrill, P.H.; Nayyar, N.; Gill, C.M.; et al. Germline and Somatic BAP1 Mutations in High-Grade Rhabdoid Meningiomas. Neuro-Oncol. 2017, 19, 535–545, doi:10.1093/neuonc/now235.
  • 52. Smith, M.J.; O’Sullivan, J.; Bhaskar, S.S.; Hadfield, K.D.; Poke, G.; Caird, J.; et al. Loss-of-Function Mutations in SMARCE1 Cause an Inherited Disorder of Multiple Spinal Meningiomas.  Genet.201345, 295–298.

  • “Mutations of the SMO and SUFU genes are observed in meningiomas of the skull base, while the TRAF7 mutation is found in meningiomas of the anterior and medial skull base [42].” It is not clear how this relates mutation to the site of origin? Also, again a review article is cited here. It would be much better to cite primary research studies than secondary reviews. Similar statement for lines 294-302.
  • In accordance with the reviewer's suggestions we have modified the text. We have also added new references.

It has recently been hypothesized that the mutations and the direct molecular pathways activated in the processes of tumorigenesis could be related to the embryological cells of origin and, therefore, to the tumor site (Tab. 3) [44,47,57]. Okano et al., demonstrated that, NF-2 mutant tumors originate from neural crest-derived arachnoid cells, while non-NF-2 tumors stem from dorsal and paraxial mesoderm [58]. AKT1 mutations are demonstrable in anterior skull base and convexity meningiomas, while SMO mutations are found in anterior skull base meningiomas [44]. NF2 mutations are frequent in intraventricular meningiomas [59]. A large genetic analysis found high rates of NF-2 and POLR2A alterations in posterior fossa meningiomas [60].

  • Clark, V.E.;Erson-Omay, E.Z.; Serin, A.; Yin, J.; Cotney, J.; Ozduman, K.; et al. Genomic Analysis of non-NF2 Meningiomas Reveals Mutations in TRAF7, KLF4, AKT1, and SMO. Science 2013, 339, 1077-1080. doi: 10.1126/science.1233009.
  • 47. Clark, V.E.; Harmanc, A.S.;, Bai. H.; et al. Recurrent Somatic Mutations in POLR2A Define a Distinct Subset of Meningiomas. Nat Genet. 2016, 48, 1253–1259.
  • 57. Boetto, J.; Bielle, F.; Sanson, M.; Peyre, M.; Kalamarides, M. SMO Mutation Status Defines a Distinct and Frequent Molecular Subgroup in Olfactory Groove Meningiomas. Oncol. 2017, 19, 345-351
  • 58. Okano, A.; Miyawaki, S.; Hongo, H.; Dofuku, S.; Teranishi, Y.; Mitsui, J.; et al. Associations of Pathological Diagnosis and Genetic Abnormalities in Meningiomas with the Embryological Origins of the Meninges. Rep. 2021, 11, 6987.
  • 59. Jungwirth, G.; Warta, R.; Beynon, C.; Sahm, F.; von Deimling, A.; Unterberg, A.; et al. Intraventricular Meningiomas Frequently Harbor NF2 Mutations but Lack Common Genetic Alterations in TRAF7, AKT1, SMO, KLF4, PIK3CA, and TERT. Acta Neuropathol. Commun. 2019, 7, 140. doi: 10.1186/s40478-019-0793-4.
  • 60. Youngblood, M.W.; Duran, D.; Montejo, J.D.; Li, C.; Omay, S.B.; Ozduman, K.; et al. Correlations Between Genomic Subgroup and Clinical Features in a Cohort of More Than 3000 Meningiomas. Neurosurg. 2019, 133, 1345–1354.

  • "In grade 2 meningiomas is frequent the mutation of SMARCE1 (97%)" This appears to state that 97% of grade 2 meningiomas have SMARCE1 mutations, which is not accurate.
  • We thank you for your correct observation. We have, accordingly, corrected the sentence and added pertinent references. “In clear cell meningioma (WHO grade II) is frequent the mutation of SMARCE1 [61-62].”
  • 61. Sievers, P.; Sill, M.; Blume, C.; Tauziede‑Espariat, A.; Schrimpf, D.; Stichel, D.; et al. Clear Cell Meningiomas are Defined by a Highly Distinct DNA Methylation Profile and Mutations in SMARCE1. Acta Neuropathol. 2021, 141, 281–290. org/10.1007/s00401-020-02247-2.
  • 62. Tauziede-Espariat, A.; Parfait, B.; Besnard, A.; Lacombe, J.; Pallud, J.; Tazi, S.; et al. Loss of SMARCE1 Expression is a Specific Diagnostic Marker of Clear Cell Meningioma: a Comprehensive Immunophenotypical and Molecular Analysis. Brain Pathol. 2018, 28, 466–474. doi:10.1111/bpa.12524.

  • Lines 324 – 330: there are several places where semi-colons are used instead of periods.
  • We have corrected the text.

  • For the section “4.1. Targeted Therapy” it might improve readability of the single large block of text was broken into paragraphs for each of the drugs discussed.
  • As you suggested, we have divided the section 4.1 “Targeted Therapy” into several paragraphs.

  • “Interferon-alpha is a biological agent that demonstrates a slight therapeutic benefit in recurrent meningiomas.” There should be a reference for this.
  • We have added the relevant reference.
  • 99. Chamberlain, M.C. IFN-α for Recurrent Surgery- and Radiation-Refractory High-Grade Meningioma: a Retrospective Case Series. CNS Oncol. 2013, 2, 227-235. Doi:10.2217/cns.13.17.

  • “slight improvement in PFS6, with” Does this refer to PFS at 6 months? This should be defined.
  • We have made the appropriate corrections.

  • “little research has been done to identify new and alternative treatment modalities". This statement is not accurate. You could say that no other therapies besides radiation/surgery have been found to be effective with conclusive evidence.
  • The sentence has been removed.

  • “However, very little is known about the pathogenesis of meningiomas and the molecular alterations promoting meningioma proliferation.” As discussed above, this is not accurate.
  • This sentence has been removed.

  • The discussion of micro-RNAs should be moved to the main text (not the conclusions)
  • As you suggested, we have removed the microRNAs from the "Conclusions" section.

Comments on the Quality of English Language

  • Only minor issues in understanding the text, however there are many areas where the syntax/grammar could be improved.
  • The manuscript, as advised by the reviewer, has undergone a careful review of the English language and syntax/grammar.
